

# Assessment of three risk evaluation systems for patients aged ≥70 in East China: performance of SinoSCORE, EuroSCORE II and the STS risk evaluation system

Lingtong Shan[1,*], Wen Ge[2,*], Yiwei Pu[1,*], Hong Cheng[3,*], Zhengqiang Cang[9], Xing Zhang[10], Qifan Li[1], Anyang Xu[4], Qi Wang[10], Chang Gu[5] and Yangyang Zhang[6,7,8]

[1] The First Clinical Medical College, Nanjing Medical University, Nanjing, China

[2] Department of Cardiothoracic Surgery, Shuguang Hospital affiliated to Shanghai University of TCM, Shanghai, China

[3] Department of Neurology, Jiangsu Province People's Hospital, The First Affiliated Hospital of Nanjing Medical University, Nanjing, China

[4] Department of Chronic and Noncommunicable Disease, Shanghai Changning District Center for Disease Control and Prevention, Shanghai, China

[5] Department of Thoracic Surgery, Shanghai Chest Hospital, Shanghai JiaoTong University, Shanghai, China

[6] Key Laboratory of Arrhythmias of the Ministry of Education of China, East Hospital, Tongji University School of Medicine, Shanghai, China

[7] Department of Cardiovascular Surgery, East Hospital, Tongji University School of Medicine, Shanghai, China

[8] Department of Thoracic and Cardiovascular Surgery, Jiangsu Province People's Hospital, The First Affiliated Hospital of Nanjing Medical University, Nanjing, China

[9] Department of Burn and Plastic Surgery, Jinling Hospital, Nanjing Medical University, Nanjing, China

[10] The Fourth Clinical Medical College, Nanjing Medical University, Nanjing, China

[*] These authors contributed equally to this work.

Corresponding author
Yangyang Zhang,
zhangyangyang_wy@vip.sina.com

## ABSTRACT

**Objectives.** To assess and compare the predictive ability of three risk evaluation systems (SinoSCORE, EuroSCORE II and the STS risk evaluation system) in patients aged ≥70, and who underwent coronary artery bypass grafting (CABG) in East China.

**Methods.** Three risk evaluation systems were applied to 1,946 consecutive patients who underwent isolated CABG from January 2004 to September 2016 in two hospitals. Patients were divided into two subsets according to their age: elderly group (age ≥70) with a younger group (age <70) used for comparison. The outcome of interest in this study was in-hospital mortality. The entire cohort and subsets of patients were analyzed. The calibration and discrimination in total and in subsets were assessed by the Hosmer–Lemeshow and the C statistics respectively.

**Results.** Institutional overall mortality was 2.52%. The expected mortality rates of SinoSCORE, EuroSCORE II and the STS risk evaluation system were 0.78(0.64)%, 1.43(1.14)% and 0.78(0.77)%, respectively. SinoSCORE achieved the best discrimination (the area under the receiver operating characteristic curve (AUC) = 0.829), followed by the STS risk evaluation system (AUC = 0.790) and EuroSCORE II (AUC = 0.769) in the entire cohort. In the elderly group, the observed mortality rate was 4.82% while it was 1.38% in the younger group. SinoSCORE (AUC = .829) also achieved

the best discrimination in the elderly group, followed by the STS risk evaluation system (AUC = .730) and EuroSCORE II (AUC = 0.640) while all three risk evaluation systems all had good performances in the younger group. SinoSCORE, EuroSCORE II and the STS risk evaluation system all achieved positive calibrations in the entire cohort and subsets.

**Conclusion**. The performance of the three risk evaluation systems was not ideal in the entire cohort. In the elderly group, SinoSCORE appeared to achieve better predictive efficiency than EuroSCORE II and the STS risk evaluation system.

## INTRODUCTION

The number of cardiac surgeries in China is increasing steadily, especially coronary artery bypass grafting (CABG) (*Wang et al., 2016*). Cardiac surgeries are currently regarded as safe and effective owing to the development of surgical, anesthetic and perioperative management (*Hu et al., 2016*). Preoperative risk evaluation systems play an important role in current cardiac surgical practice. During the last two decades, various risk evaluation systems have been developed to predict mortality for cardiac surgery, such as the European System for Cardiac Operative Risk Evaluation (EuroSCORE) (*Nashef et al., 1999*; *Chalmers et al., 2013*) and the Society of Thoracic Surgeons (STS) score (*Fortescue, Kahn & Bates, 2001*). In China, Sino System for Coronary Operative Risk Evaluation (SinoSCORE) (*Li, Zheng & Hu, 2009*; *Zheng & Zhang, 2010*), based on more than 9,000 Chinese patients was published in 2010.

Previous studies only evaluated the predictive capacity of these risk evaluation systems among different operations and had rarely evaluated the ability to predict mortality among different age groups (*Collins & Manach, 2016*; *Allyn et al., 2017*; *Aggarwal et al., 2016*; *Ad et al., 2016*; *Garcia-Valentin et al., 2016*). Therefore, our study attempts to analyze the predictive capacity of the three risk evaluation systems for patients aged ≥70 who were treated with isolated CABG operation in East China.

## METHODS

### Patients

Form January 2004 to September 2016, 2,070 patients from two hospitals (the First Affiliated Hospital of Nanjing Medical University and the East Hospital affiliated to Tongji University) who underwent isolated CABG were invited to participate in the study, which was approved by ethic committees of the two hospitals (Ethical Application Ref: 2017-SR-053; [2017] research (018), respectively). All patients had been discharged when data was extracted. Inclusion criteria were isolated CABG while exclusion criteria were CABG combined with other cardiac surgeries and missing of information. All enrolled patients signed informed consent forms. There were 124 (5.99%) patients excluded

from the analysis because of incomplete data ($n = 84$, 4.01%) and participation in developing SinoSCORE ($n = 41, 1.98\%$), and a total of 1,946 procedures constituted made up the database. Each patient's diagnosis was confirmed by coronary arteriography. According to the study database, the operative risk was predicted by the algorithms online: SinoSCORE is available at http://www.cvs-china.com/sino.asp, EuroSCORE II is available at http://www.euroscore.org/calc.html and the STS risk evaluation system is available at http://riskcalc.sts.org/STSWebRiskCalc273/de.aspx. The expected mortality rates of each patient were ascertained by every system above.

The average life expectancy of the Chinese people has reached 76.34 years old, while the mortality rate of CAD has increased significantly in people over 70 years old. To explore the predictive efficacy of the three evaluation systems, in each set, patients were divided into two subsets according to their age: elderly group (age $\geq 70$, 33.04%, 643/1,946) and younger group (age<70, 66.96%, 1,303/1,946). The calibration and discrimination of the three systems in entire cohort and each subset were assessed and compared. Furthermore, in order to make a fair comparison among the three systems, we compared the expected and observed mortality rates in entire cohort and each subset, respectively.

## Outcome endpoint

The outcome of interest in this study was in-hospital mortality, which was defined as postoperative in-hospital all-cause death.

## Statistical analysis

Statistical analysis was performed with SPSS version 19.0 (SPSS Inc., Chicago, IL, USA). If continuous variables satisfy the normal distribution, then variables were expressed as mean $\pm$ standard deviation, else variables were expressed as median and interquartile range (IQR). Categorical variables were expressed as percentages. Statistical analysis comparing groups was performed using $t$-tests for continuous variables, Mann–Whitney–Wilcoxon tests for ordinal or continuous variables that do not satisfy the assumptions required for a $t$-test, and Fisher's exact or $\chi^2$ (chi-square) tests for categorical variables. A $P$-value of less than 0.05 was considered significant.

Performance of the three risk evaluation systems was assessed by comparing the expected and observed in-hospital mortality rates. Calibration (statistical precision) was analyzed by Hosmer–Lemeshow (H-L) goodness-of-fit statistic (*Lemeshow & Hosmer Jr, 1982*). The H-L statistic measured the differences between expected and observed outcomes. $P$-value greater than 0.05 means there is no evidence that this risk evaluation system is poorly calibrated. The area under the receiver operating characteristic curve (AUC), which was used to assess how well the evaluated system could discriminate between survivors and non-survivors, describes an estimate of the model's discrimination ability (*Hanley & Mcneil, 1982*). The discriminative power of the risk evaluation system is considered excellent if the AUC >0.80, good if >0.75 and acceptable if >0.70. Calibration plots of observed versus expected mortality rates for 20 equally sized groups by ranked expected rates calculated of the three systems were constructed. The ideal calibrated predictions consist of a 45° line, Where points below or above the diagonal line indicated overestimation or

underestimation respectively. Plots of observed and expected mortality rates of the three systems by age distribution were also drawn, to illustrate the differences mortality rate in subsets.

Finally, the net benefit of three risk evaluation systems for predicting in-hospital mortality was performed by Decision Curve Analysis (DCA). DCA consisted in the subtraction of the proportion of all patients who are false-positive from the proportion who are true-positive, weighting by the relative harm of a false-positive and a false-negative result. DCA was performed using R software version 3.4.0 (*R Core Team, 2017*) with the package Decision curve.

## RESULTS

### Performance in elderly group

There were 31 observed deaths, with an observed mortality rate of 4.82% (Table 1). Expected mortality rate of SinoSCORE, EuroSCORE II and the STS risk evaluation system for the elderly group were 1.06(0.87)%, 2.21(1.29)% and 1.27(0.97)%, respectively. The expected ability of the three risk evaluation systems in the elderly group is shown in Table 2. SinoSCORE, EuroSCORE II and the STS risk evaluation system all showed positive calibration in predicting in-hospital mortality (H-L: $P = 0.053$, $P = 0.389$ and $P = 0.061$, respectively) (Table 2). The discriminative power for the elderly group in SinoSCORE was the best (0.829), followed by the STS risk evaluation system (0.730) and EuroSCORE II (0.640) (Fig. 1). SinoSCORE showed excellent discrimination in elderly patients while EuroSCORE II had poor discrimination in elderly patients. The decision curve analysis (DCA) showed the clinical practicability of SinoSCORE, EuroSCORE II and the STS risk evaluation system when the clinician predicted the mortality of patients after cardiac surgery. The results were presented as a graph with the selected probability threshold (i.e., the degree of certitude of postoperative mortality over which patients refused operation) plotted on the abscissa and the net benefits of the risk evaluation system on the ordinate. The net benefit of the SinoSCORE was always greater than that of EuroSCORE II and the STS risk evaluation system regardless of the selected threshold, included between 0 and 20% (Fig. 2).

### Performance in younger group

There were 18 observed deaths, with an observed mortality rate of 1.38%. Expected mortality rates of SinoSCORE, EuroSCORE II and the STS risk evaluation system for the younger group were 0.67(0.50)%, 1.18(0.70)% and 0.60(0.50)%, respectively. The expected ability of the three risk evaluation systems in the younger group is shown in Table 2. SinoSCORE, EuroSCORE II and the STS risk evaluation system showed positive calibration (H-L: $P = 0.643$, $P = 0.527$ and $P = 0.321$, respectively) (Table 2). The discriminative power for the younger group in EuroSCORE II was the highest (0.785), followed by the STS risk evaluation system (0.772) and SinoSCORE (0.769) (Table 2, Fig. 1). Three risk evaluation systems all showed good discrimination in the younger group. The decision curve of SinoSCORE, EuroSCORE II and the STS risk evaluation system remained very close regardless of the threshold selected. The benefit of EuroSCORE II was relatively

**Table 1  Baseline clinical characteristics of subgroups.**

| Risk factors | Elderly group (n = 643) | Younger group (n = 1,303) | P-value |
|---|---|---|---|
| Age (y) | 74.00(5.00) | 62.00(8.00) | <0.001 |
| Female (n, %) | 135(20.96) | 261(20.03) | 0.619 |
| Weight (kg) | 67.00(13.00) | 70.00(14.00) | <0.001 |
| Height (cm) | 168.00(11.00) | 168.00(10.00) | 0.021 |
| BMI (kg/m$^2$) | 24.22(3.79) | 25.00(3.90) | <0.001 |
| Morbid obesity (n, %) | 22(3.42) | 76(5.83) | 0.022 |
| Body surface area (m$^2$) | 1.72 ± 0.15 | 1.76 ± 0.16 | <0.001 |
| Diabetes (n, %) | 196(30.48) | 395(30.31) | 0.940 |
| Hypertension (n, %) | 454(70.61) | 868(66.62) | 0.076 |
| Renal failure (n, %) | 9(1.40) | 11(0.84) | 0.253 |
| Serum creatinine (μmol/l) | 82.00(27.30) | 74.90(24.00) | <0.001 |
| Ccr (ml/min) | 64.16(23.18) | 86.57(32.07) | <0.001 |
| Stroke (n, %) | 18(2.72) | 22(1.66) | 0.104 |
| COPD (n, %) | 24(3.73) | 19(1.46) | 0.001 |
| Peripheral vascular disease (n, %) | 18(2.80) | 25(1.92) | 0.214 |
| Previous cardiac surgery (n, %) | 27(4.20) | 42(3.22) | 0.274 |
| Atrial flutter and fibrillation (n, %) | 17(2.64) | 23(1.77) | 0.199 |
| Pulmonary hypertension (n, %) | 56(8.71) | 100(7.67) | 0.728 |
| Myocardial infarction (n, %) | 74(11.21) | 180(13.56) | 0.167 |
| Unstable angina pectoris (n, %) | 388(58.79) | 726(54.71) | 0.096 |
| Number of (n, deseased vessels (n) | 3.00(0.00) | 3.00(0.00) | 0.082 |
| Three-vessel coronary disease (n, %) | 598(90.60) | 1,167(87.94) | 0.101 |
| NYHA IV (n, %) | 20(3.03) | 17(1.28) | 0.013 |
| LVEF (%) | 63.00(5.80) | 63.10(6.00) | 0.425 |
| Preoperative IABP (n, %) | 10(1.56) | 5(0.38) | 0.040 |
| Status of surgery | | | 0.999 |
|    Elective (n, %) | 611(95.02) | 1,238(95.01) | |
|    Urgent (n, %) | 25(3.89) | 51(3.91) | |
|    Salvage (n, %) | 7(1.09) | 14(1.07) | |
| Number of grafts (n) | 3.00(1.00) | 4.00(1.00) | <0.001 |
| Hospital mortality (n, %) | 31(4.82) | 18(1.38) | <0.001 |

**Notes.**
Abbreviations: BMI, body mass index; COPD, chronic obstructive pulmonary disease; Scr, Serum creatinine; Ccr, endogenous creatinine clearance rate; LVEF, left ventricular ejection fraction.

greater than that of SinoSCORE and the STS risk evaluation system: between 0 and 20% (Fig. 2).

## Patient baseline data

Of the 1,946 patients, the median age was 66.00(11.00) years, 643 (33.04%) were aged ≥70 years old, 396 (20.35%) were female, and median left ventricular ejection fraction (LVEF) was 63.00 (6.00)%. There were 49 observed deaths, with an overall observed mortality rate of 2.52% (Table 3). The elderly group was more likely to have higher serum creatinine

**Table 2  Observed and expected mortality rates of the three systems.**

| | Elderly group of SinoSCORE | Younger group of SinoSCORE | Total patients of SinoSCORE | Elderly group of EuroSCORE II | Younger group of EuroSCORE II | Total patients of EuroSCORE II | Elderly group of STS | Younger group of STS | Total patients of STS |
|---|---|---|---|---|---|---|---|---|---|
| Number of patients | 643 | 1,303 | 1,946 | 643 | 1,303 | 1,946 | 643 | 1,303 | 1,946 |
| Deaths | 31 | 18 | 49 | 31 | 18 | 49 | 31 | 18 | 49 |
| Observed mortality (%) | 4.82 | 1.38 | 2.52 | 4.82 | 1.38 | 2.52 | 4.82 | 1.38 | 2.52 |
| Expected mortality (%) | 1.06(0.87) | 0.67(0.50) | 0.78(0.64) | 2.21(1.29) | 1.18(0.70) | 1.43(1.14) | 1.27(0.97) | 0.60(0.50) | 0.78(0.77) |
| AUC | 0.829 | 0.769 | 0.829 | 0.640 | 0.785 | 0.769 | 0.730 | 0.772 | 0.790 |
| H–L statistics | 0.053 | 0.643 | 0.411 | 0.389 | 0.527 | 0.113 | 0.061 | 0.321 | 0.230 |

**Notes.**

Abbreviations: AUC, area under receiver operating characteristic curve; H–L statistics, Hosmer-Lemeshow statistics.

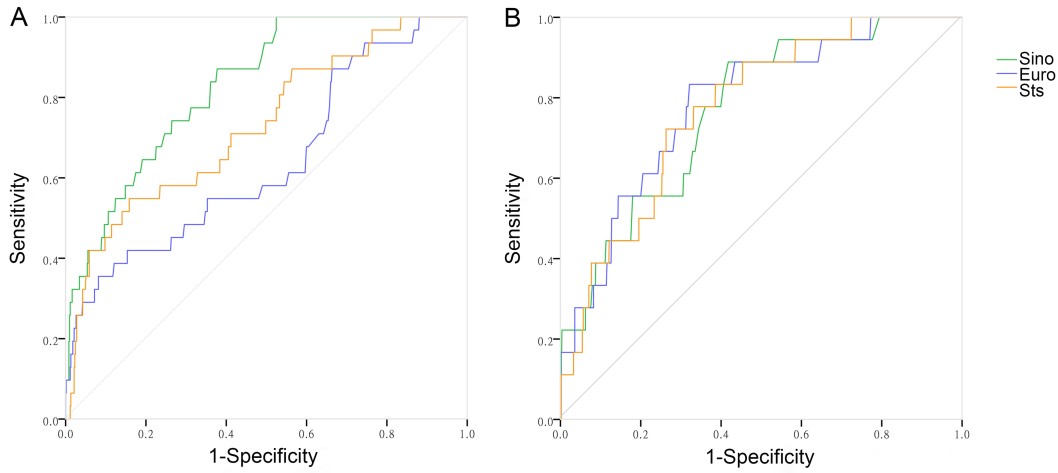

**Figure 1  The receiver operating characteristic curves of the three risk evaluation systems with subsets.**
(A) The receiver operating characteristic curves of the three risk evaluation systems with the elderly group
(SinoSCORE 0.829, EuroSCORE II 0.640 and STS 0.730). (B) The receiver operating characteristic curves
of the three risk evaluation systems with the younger group (SinoSCORE 0.769, EuroSCORE II 0.785 and
STS 0.772).

($P < 0.001$), chronic obstructive pulmonary disease (COPD) ($P = 0.001$), The New
York Heart Association (NYHA) class IV ($P = 0.013$), preoperative intra-aortic balloon
pump (IABP) ($P = 0.040$), and hospital mortality ($P < 0.001$). The younger group had
higher weight ($P < 0.001$), body mass index ($P < 0.001$), body surface area ($P < 0.001$),
endogenous creatinine clearance rate ($P < 0.001$), proportionately more grafts ($P < 0.001$)
and number of grafts ($P < 0.001$) (Table 1).

Expected mortality mean mortality percentage of SinoSCORE, EuroSCORE II and
the STS risk evaluation system for the entire cohort were 0.78(0.64)%, 1.43(1.14)% and
0.78(0.77)%, respectively. The expected abilities of the three risk evaluation systems
in entire cohort were shown in Table 2. SinoSCORE, EuroSCORE II and the STS risk
evaluation system all showed positive calibration in predicting in-hospital mortality (H-L:
$P = 0.411$, $P = 0.113$ and $P = 0.230$, respectively) (Table 2). The discriminative power
for the entire cohort (AUC) in SinoSCORE was the best (0.829), followed by the STS risk
evaluation system (0.790) and EuroSCORE II (0.769) (Fig. 3).

Calibration plots showed that the three risk evaluation systems deviated from the
diagonal line, so these risk evaluation systems all underestimated mortality rates in the
entire cohort (Fig. 4). When the age factor was considered, the mortality obviously increased
with age. SinoSCORE, EuroSCORE II and the STS risk evaluation system overestimated
in-hospital mortality rates in patients under 55 years, but underestimated in-hospital
mortality rates in patients over 55 years (Fig. 5).

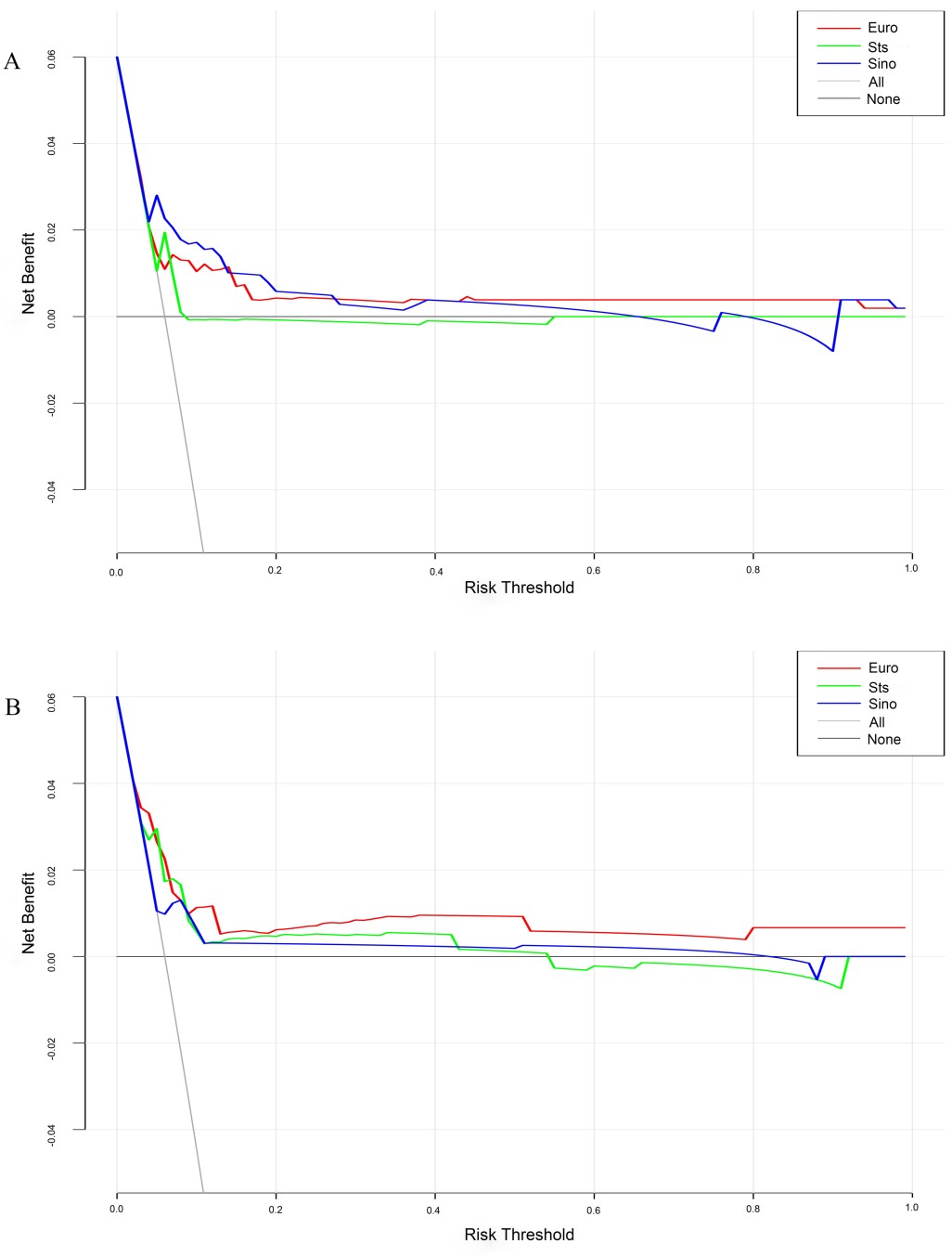

**Figure 2** **DCA showed the clinical usefulness of SinoSCORE, EuroSCORE II and the STS risk evaluation system in predicting in-hospital mortality.** The gray line represents the net benefit of providing surgery for all patients, assuming that all patients would survive. The black line represents the net benefit of surgery to no patients, assuming that none would survive after surgery. The red, blue and green lines represent the net benefit of applying surgery to patients according to EuroSCORE II, SinoSCORE, and the STS risk evaluation system, respectively. The selected probability threshold is plotted on the abscissa. (A) DCA for the elderly group; (B) DCA for the younger group.

**Table 3  Cardiac surgery patient baseline clinical characteristics.**

| Risk factors | Total ($n = 1,946$) |
|---|---|
| Age (y) | 66.00(11.00) |
| Female ($n$, %) | 396(20.35) |
| Weight (kg) | 70.00(13.00) |
| Height (cm) | 168.00(10.00) |
| BMI (kg/m$^2$) | 24.80(3.81) |
| Morbid obesity ($n$, %) | 98(5.04) |
| Body surface area (m$^2$) | $1.75 \pm 0.16$ |
| Diabetes ($n$, %) | 591(30.37) |
| Hypertension ($n$, %) | 1,322(67.93) |
| Renal failure ($n$, %) | 20(1.03) |
| Serum creatinine ($\mu$mol/l) | 76.70(25.90) |
| Ccr (ml/min) | 78.47(33.34) |
| Stroke ($n$, %) | 40(2.06) |
| COPD ($n$, %) | 43(2.21) |
| Peripheral vascular disease ($n$, %) | 43(2.21) |
| Previous cardiac surgery ($n$, %) | 69(3.55) |
| Atrial flutter and fibrillation ($n$, %) | 40(2.06) |
| Pulmonary hypertension ($n$, %) | 156(8.02) |
| Myocardial infarction ($n$, %) | 250(12.85) |
| Unstable angina pectoris ($n$, %) | 1,086(55.81) |
| Number of diseased vessels ($n$) | 3.00(0.00) |
| Three-vessel coronary disease ($n$, %) | 1,729(88.85) |
| NYHA IV ($n$, %) | 34(1.75) |
| LVEF (%) | 63.00(6.00) |
| Preoperative IABP ($n$, %) | 43(2.21) |
| Status of surgery | |
|    Elective ($n$, %) | 1,849(95.02) |
|    Urgent ($n$, %) | 76(3.91) |
|    Salvage ($n$, %) | 21(1.08) |
| Number of grafts ($n$) | 3.00(1.00) |
| Hospital mortality ($n$, %) | 49(2.52) |

**Notes.**

Abbreviations: BMI, body mass index; COPD, chronic obstructive pulmonary disease; Scr, Serum creatinine; Ccr, endogenous creatinine clearance rate; LVEF, left ventricular ejection fraction.

## DISCUSSION

Preoperative risk assessment of patients undergoing CABG is critical for treatment decisions-making, prognostic judgments, preoperative patient education and quality-assurance measure (*Patratdelon et al., 2016*). The EuroSCORE II and the STS risk evaluation system, which were widely used around the world, have been well evaluated in different countries and regions (*Ad et al., 2016*; *Aydın et al., 2015*; *Holinski et al., 2015*; *George et al., 2015*). However, they may not accurately predict preoperative risk of Chinese patients and they require validation before clinical application (*Bai et al., 2016*; *Wang et al., 2014*). In

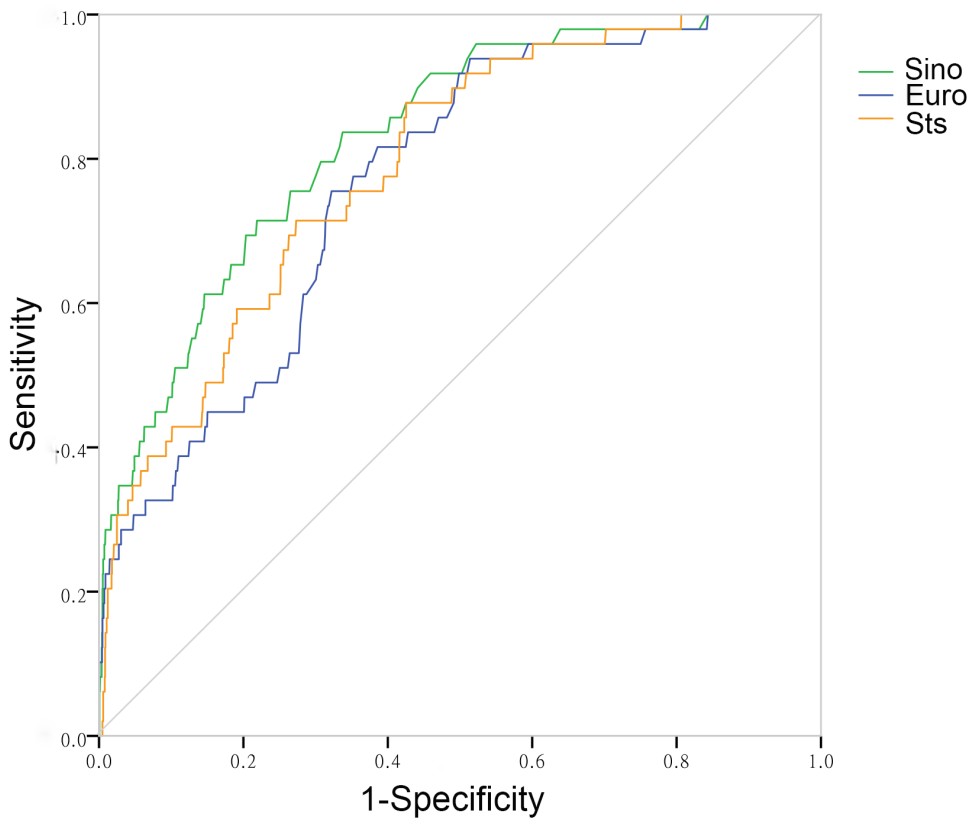

**Figure 3** **The receiver operating characteristic curves of three risk evaluation systems in total patients.** For all total patients, the receiver operating characteristic curve of SinoSCORE is 0.829, EuroSCORE is 0.769, and of the STS risk evaluation system is 0.790, respectively.

China, based on more than 9,000 patients in a national multi-centre database known as the Chinese Coronary Artery Bypass Grafting Registry Study (*Li, Zheng & Hu, 2009*; *Zheng, Li & Hu, 2009*), Sino System for Coronary Operative Risk Evaluation (SinoSCORE) was published in 2010 (*Zheng & Zhang, 2010*). After its publication, SinoSCORE was accepted and achieved good assessments in several medical centres in China (*Yu et al., 2015*; *Feng et al., 2014*; *Liu et al., 2013*).

Patients from the two regional central hospitals in East China could represent typical East China patients. Additionally, there was no difference in surgical techniques. Any risk evaluation system could show its best performance only when the patient's characteristics and treatment process were similar to those which the system was originated. Therefore, risk evaluation system should be tested in the local population before used reliably. So it is very meaningful to assess and compare the predictive ability of the three risk evaluation systems in East China patients, especially in the elderly. However, previous studies rarely evaluated the ability to predict mortality among subsets grouped by age.

In this study, SinoSCORE was superior to EuroSCORE II and the STS risk evaluation system in predicting operative mortality both in entire cohort and in the elderly group. There are some possible reasons to explain the results: (1) SinoSCORE was based on

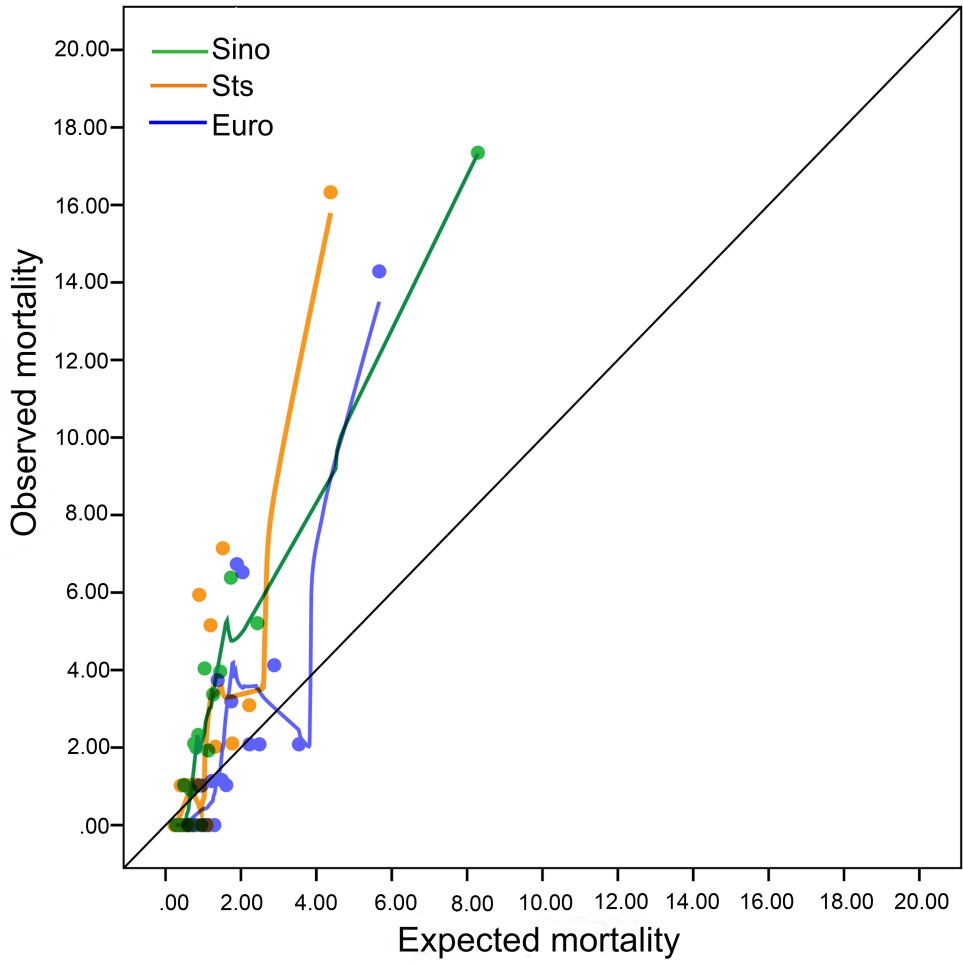

**Figure 4** **Calibration plots for the three risk evaluation systems.** Calibration plots for SinoSCORE, EuroCORE and STS.

Chinese patients, the characteristics of patients were similar to the study database; (2) Removing 41 cases involved in SinoSCORE, we compared the differences between our local database and the SinoSCORE database. There were some similar risk factors between two databases, such as age, hypertension, diabetes, renal failure, peripheral vascular disease, active endocarditis, critical preoperative state, three-vessel coronary disease and in-hospital mortality (*Li, Zheng & Hu, 2009*; *Zheng & Zhang, 2010*; *Zheng, Li & Hu, 2009*), and which might be the reason contributed to SinoSCORE's a good expected power (Table 4); (3) SinoSCORE was designed for isolated CABG operation, while EuroSCORE II and the STS risk evaluation system were not only suitable for isolated CABG but also for other cardiac surgeries, such as valvular surgery and aortic surgery. Therefore, SinoSCORE was more suitable in the study and showed better expected performance.

Although the three systems all had positive calibration and discrimination, it was a pity that they substantially underestimated the mortality in the entire cohort and subsets. The discrepancy between observed and expected mortality was particularly high in the elderly

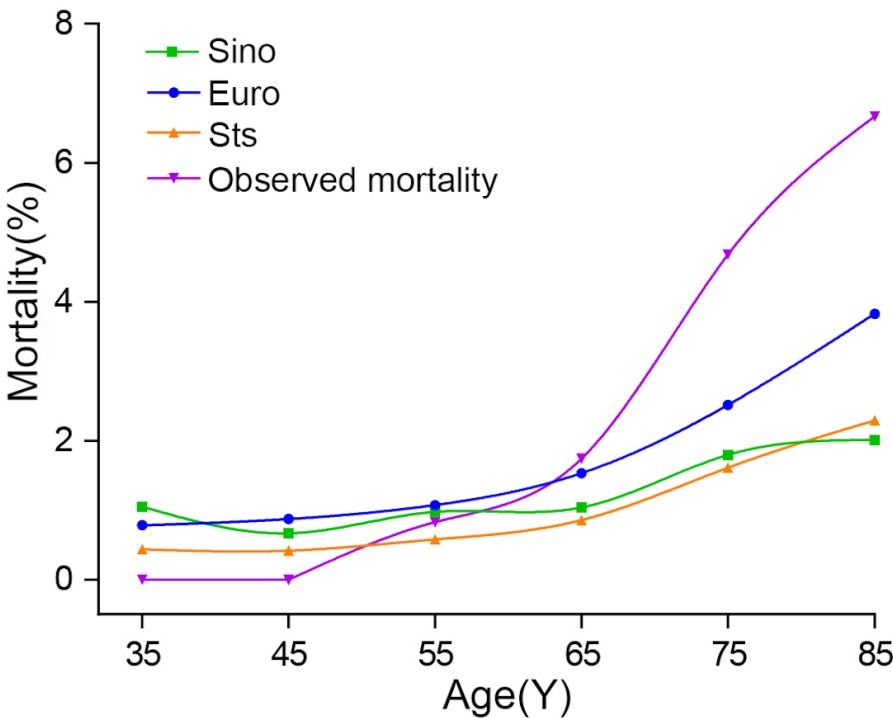

**Figure 5** **Plots of observed and expected mortality rates of the three systems by age distribution.** Three risk evaluation systems overestimated in-hospital mortality rates in patients under 55 years, but underestimated in-hospital mortality rates in patients over 55 years.

group (4.82% vs.1.06%, 2.21%, 1.27%, respectively). One reason may be related to the predictive accuracy of the three systems; another is that the preoperative parameters of the elderly were different from those of normal population.

Aging is a global problem; China, as a developing country, is also facing this problem. However, compared with some developed countries, China's aging is not so serious (*Wu et al., 2015*; *Li et al., 2015*; *Zhen et al., 2015*). In this study, patients over 70 years old accounted for 33.04%. Previous literature has reported that the proportion of patients over 70 years was up to 60% (*Churpek et al., 2015*). Patients are now older, have more comorbidities and more coronary lesions with decreased cardiac function. Some risk factors such as hypertension, COPD, NYHA class IV, preoperative IABP and more diseased vessels are more common in elderly patients. It is generally known that the risk of cardiac surgery in elderly patients is higher than that of the general population. Thus, preoperative assessment of surgical risk is more important for the elderly.

The STS risk evaluation system was widely used and has been found to be effective for predicting mortality in short and long term. EuroSCORE II is another system which was widely accepted and implemented not only in Europe, North America but also in Asia. EuroSCORE II included fewer variables than the STS risk evaluation system (18 vs 26), thereby making it easier to use in clinical practices. However, SinoSCORE has only 11 variables, whose weights were obviously different from the previous two systems.

**Table 4** The baseline of SinoSCORE database and Local database.

| Risk factors | SinoSCORE (N = 9,248) | Local (N = 1,946) |
|---|---|---|
| Age (y) | 62.60 ± 9.20 | 65.28 ± 8.33 |
| Female (%) | 21.50 | 20.35 |
| Diabetes (%) | 26.40 | 30.37 |
| Hypertension (%) | 63.50 | 67.93 |
| Renal failure (%) | 0.60 | 1.03 |
| Cerebrovascular accident (%) | 8.30 | 2.06 |
| COPD (%) | 1.30 | 2.21 |
| Peripheral vascular disease (%) | 2.50 | 2.21 |
| Previous cardiac surgery (%) | 2.30 | 3.55 |
| Active endocarditis (%) | 0 | 0 |
| Critical preoperative state (%) | 4.60 | 5.50 |
| Myocardial infarction (%) | 9.60 | 12.85 |
| Unstable angina pectoris (%) | 31.10 | 55.81 |
| Three-vessel coronary disease (%) | 76.70 | 88.85 |
| Emergency (%) | 7.10 | 4.98 |
| Pulmonary hypertension (%) | 1.10 | 8.02 |
| LVEF 30–50% (%) | 20.90 | 8.70 |
| LVEF <30% (%) | 0.90 | 0.20 |
| Isolated CABG(%) | 87.80 | 100 |
| Hospital mortality (%) | 3.27 | 2.52 |

**Notes.**
Abbreviations: COPD, chronic obstructive pulmonary disease; LVEF, left ventricular ejection fraction.

Comparing the variables of the three systems, only a few were the same, such as age, diabetes, status of surgery and type of surgery, which could lead to the difference in predictive capabilities among them.

Previous literatures reported the in-hospital mortality rate of patients undergoing CABG in developed countries was 2.18~2.50% (*Moazzami et al., 2017*; *Curtis et al., 2017*; *Swaminathan et al., 2016*; *Herlitz et al., 2015*; *Kuwaki et al., 2015*). The mortality rate in this study was 2.52%, slightly higher than previous reported. It also explained that the three risk evaluation systems under estimated mortality rates. One possible reason was that although cardiac surgery and perioperative care in China had developed rapidly in the last decades, there were still some gaps compared with the developed countries. Another possible reason was that there were 124 patients excluded from the study. The quality control of the hospital required all in-hospital deaths should be archived. As a result, this study included all deaths (except two death developing SinoSCORE); to some extent, the mortality rate had increased.

SinoSCORE solved the problem that China did not have its own heart surgery risk evaluation system. Although it was published recently, SinoSCORE has achieved good assessments in this study, as it did in other reports (*Yu et al., 2015*; *Feng et al., 2014*; *Liu et al., 2013*). When compared with EuroSCORE II and the STS risk evaluation system, SinoSCORE shows no compromise.

SinoSCORE was designed to evaluate isolated CABG mortality, where as EuroSCORE II and the STS risk evaluation system can be used to predict other cardiac surgical mortality. Also, the STS risk evaluation system can also predict other outcomes, evaluating the predictive capacities of the STS risk evaluation system to predict only operative mortality may undermine its potency. This study was a double-centre retrospective and non-randomized observational study. The sample size was still small compared with other studies that were sourced from a large number of cases. The three risk evaluation systems underestimated the mortality in the entire corhort and subsets in spite of their positive discrimination and calibration. These factors might result in bias. Therefore, the mortality statistics may be limited to some degree.

## CONCLUSION

The performance of the three risk evaluation systems was not ideal, although the three risk evaluation systems showed positive discrimination and calibration in the entire cohort. SinoSCORE achieved slightly better predictive efficiency than STS risk evaluation system in elderly patients underwent CABG in East China.

## ACKNOWLEDGEMENTS

We wish to thank the help in surgeries given by Prof. Yongfeng Shao, Prof. Xiaowei Wang, Dr. Lei Wei, Dr. Xiangxiang Zheng, Dr. Haoliang Sun, Dr. Luyao Ma, and Dr. Wei Zhang.

### Funding

This work was supported by Six major talent Summit of Jiangsu Province (2015-WSW-019 to Yangyang Zhang) and the practice innovation training program projects for the Jiangsu students in college (201610312011Z to Lingtong Shan) (201510312073Z to Zhengqiang Cang). The funders had no role in study design, data collection and analysis, decision to publish, or preparation of the manuscript.

### Grant Disclosures

The following grant information was disclosed by the authors:
Six major talent Summit of Jiangsu Province: 2015-WSW-019.
Practice innovation training program projects: 201610312011Z, 201510312073Z.

### Competing Interests

The authors declare there are no competing interests.

### Author Contributions

- Lingtong Shan, Yiwei Pu, Xing Zhang and Qifan Li performed the experiments, prepared figures and/or tables, authored or reviewed drafts of the paper, approved the final draft.
- Wen Ge, Hong Cheng, Zhengqiang Cang and Qi Wang performed the experiments, authored or reviewed drafts of the paper, approved the final draft.

- Anyang Xu performed the experiments, analyzed the data, authored or reviewed drafts of the paper, approved the final draft.
- Chang Gu conceived and designed the experiments, performed the experiments, analyzed the data, contributed reagents/materials/analysis tools, authored or reviewed drafts of the paper, approved the final draft.
- Yangyang Zhang conceived and designed the experiments, contributed reagents/materials/analysis tools, authored or reviewed drafts of the paper, approved the final draft.

## Human Ethics

The following information was supplied relating to ethical approvals (i.e., approving body and any reference numbers):

The first affiliated hospital of Nanjing Medical University and the east hospital affiliated to Tongji University granted Ethical approval to carry out the study within its facilities (Ethical Application Ref: 2017-SR-053; [2017] research (018)).

## Data Availability

The raw data has been provided as a Supplemental File.

## Supplemental Information

Supplemental information for this article can be found online at http://dx.doi.org/10.7717/peerj.4413#supplemental-information.

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
