# Peer review of "Assessment of three risk evaluation systems for patients aged ≥70 in East China: performance of SinoSCORE, EuroSCORE II and the STS risk evaluation system"

_PeerJ, doi:10.7717/peerj.4413_

## Round 0.1 · original submission · Major Revisions

· Academic Editor

Major Revisions

Dear authors,

I have carefully read your paper and the comments of the reviewers, and after their analysis, I think your paper has scientific merit to be published in PeerJ, once some major issues are solved by you (see comments below). Therefore, my decision is MAJOR REVISION.

With respect and warm regards,
Dr Palazón-Bru (academic editor for PeerJ)

Reviewer 1 ·

Basic reporting

no comment

Experimental design

no comment

Validity of the findings

no comment

Additional comments

The author tried to assess and compare the predictability of three risk evaluation systems in patients aged≥70 , and who underwent coronary artery bypass grafting (CABG) in East China. Then they concluded"good discrimination and calibration in the entire cohort. In the elderly group, SinoSCORE achieved slightly better predictive efficiency than STS risk evaluation system". The manuscript can be improved significantly by performing the following:
1. Though the cohorts are large, due to the single center and single ethnicity (East China) study make it impossible to make such a strong conclusion and should explicit for limitation in the discussion section.
2. Please clarify your bias when select the population prior to your conclusion.
3. L81-82, please state why and how you classified the subgroup? Why aged 70 not others?
Your paper may be conditionally accepted after “major revision” and language polishment.

·

Basic reporting

While the text was generally clear in its intention, there were a number of typos or instances of awkward wording. Careful proof-reading and editing for style are both necessary. Some suggestions for the authors to consider in this regard are listed below under “General comments”.

Important details are sometimes omitted from the manuscript, leaving the reader to guess at what has been done. In other cases, details are provided later than I would expect; for example, the abstract’s objective and method sections both fail to mention that mortality was the endpoint of interest (I appreciate that this is almost inevitable as the outcome but the reader should know this for certain rather than expecting it to be the case). AUCs are presented in the abstract results without indication that this is what they are. The elderly group is discussed in the abstract without being defined.

Some of the details of the statistical analyses were only entirely clear after I had looked at the data, which was in a pleasibly sensible format and I thank the authors for this. These details include the reported mortality statistics which appear to be mean scores from the risk equation systems along with standard deviations (which are potentially problematic given the substantial skew in the distribution) and 95% CIs for these means (again, skew being an issue that needs to be considered).

The introduction needs to be considerably expanded to establish the context of the study and to motivate the research (experimental design). I would suggest a discussion of each of the scoring systems for the unfamiliar reader (otherwise, it is not until the discussion when the numbers of component items and some examples of these items are discussed that a reader not already familiar with the instruments is given any clue about their structure or complexity).

Also, references are made to literature looking at homogeneous patient groups and a lack of literature looking at different ages (lines 59-60), but discussion of this existing literature is lacking.

Experimental design

The knowledge gap that this manuscript aims to address should become clearer with the extra context recommended above.

I am assuming that all patients had been discharged when data was extracted but a statement to this effect (if it is not already present) would be reassuring, along with information about length of stay post-op.

I am concerned about the clearly skewed variables being described using means and standard deviations in all cases. While the arithmetic mean can be appropriate for mortality (and could also be appropriate for length of stay if this was added), the standard deviations are greatly inflated by the substantial skew in risk scores, for example. For age, perhaps median and IQR would be more useful if these are skewed as I suspect?

Related to this, no description of model diagnostics for the t-tests or the basis for deciding between Fisher’s Test and the Chi-squared tests is given (lines 93-94). Mann-Whitney-Wilcoxon would be an alternative should the assumptions of the t-test not be satisfied based on raw or transformed data.

Hosmer-Lemeshow is a test of model misfit (lines 99-100) and so a non-significant test does not indicate a well-calibrated model, merely one where there is a lack of sufficient evidence for miscalibration. The wording in the text should reflect this interpretation. Also, marginal test results (such as those on lines 129) would still concern me when there are so few events and so limited power in the test to detect departures. Note also that the test does not require deciles to be used (line 99), although this is a common choice.

Validity of the findings

The abstract states that the objectives include comparing the three systems, but no formal comparisons appear to be included beyond the inclusion of what might be a comparison between the three AUCs in the figures. Given the apparent lack of statistically significant differences in AUC between the three models (based on these p-values), the interpretation that SinoSCORE is superior in any sense beyond a simple numerical comparison of values needs to be softened (in the results and in the discussion). While the empirical estimates for AUC for SinoSCORE are indeed higher than the other systems in two of the three models, and so that might make it a reasonable choice for an organisation not already using any risk evaluation system when looking at older patients or patients in general, the lack of evidence for differences provides no strong basis for an organisation already using EuroSCORE II or STS to switch to using SinoSCORE, for example. A similar point applies to the interpretation of the DCA results (lines 151-153).

Given that all three systems underestimate mortality by what I would consider to be an unacceptable degree, I think that this suggests overall problems with the modelling of risk and I have to disagree with the statements that “good” or “positive” calibration was achieved. Given the substantial misfit from the Hosmer-Lemeshow test, I would suggest investigating the inclusion of a quadratic term, for example. Another possible cause of this misfit could be an omitted main effect, or after the addition of such, possibly one or more interaction terms. However, the sample sizes of events is small and should constrain the complexity of the statistical models, especially the age stratified models (e.g., Peduzzi, et al. using their 10 EPV recommendation).

The authors correctly identify an issue with missing data where these cases will not involve mortality. Multiple imputation would be a useful way of incorporating this data back into the analysis, correcting for this bias. This could be done as the primary analysis with the complete data as a secondary analysis, or vice versa.

The stratification by age is interesting. Given the two hospitals providing data, I would like to see this as another stratification variable. This would also help to address the issue of clustering within hospitals. If generalisation beyond these hospitals is not required, and the authors did not want to stratify by hospital, a fixed effect for the hospital could be added to logistic regression models to allow for evaluating the scores (note the limits of model complexity discussed above though). I wonder if some of the other patient characteristics could be added to age (and perhaps hospital) in this way?

The use of the same patients who were used to develop SinoSCORE (lines 188-189) is a problem as this makes the estimated performance using this instrument overly optimistic. The analysis should exclude those patients used to develop this instrument to avoid this problem and thereby provide less biased estimates of the model’s performance.

Additional comments

This manuscript examined three risk evaluation scoring systems (EuroSCORE II, STS, and SinoSCORE) as predictors of mortality from 1987 CABG patients from two hospitals in China (51 deaths occurred). The three scores are considered overall and using two age strata (<70 years, 70+ years). Note that these categories contradict the manuscript title which refers to those aged 70 and above. There is considerable misfit from each risk prediction model and it appears that no statistically significant differences were found between the three scores, although SinoSCORE was empirically better fitting according to AUC for two of the three models.

The manuscript seems well-motivated from my (a statistician's rather than a clinician’s) perspective, but the issues of poor model fit (the calibration component) and over-interpretation/lack of discussion of what appears to be no evidence of superiority of any score over the others means that the data has not been used to its full effect.

A non-exhaustive list of suggestions for improving the wording includes (with new words and/or suggested replacements in upper case):

1. “predictability” in the abstract (line 29) might be better as “predictive ability”.
2. I would delete the first “statistics” from the abstract’s methods, or make both singular (line 35).
3. Abstract: “1.51% in THE younger” (line 40), “best predictive value in THE elderly group” (line 41), “while ALL three risk evaluation systems had” (line 42).
4. Use a more precise description than “They” (line 52).
5. “systemS play an” (line 53).
6. “THE Society” (line 56).
7. “operations AND HAVE rarely evaluated their ability” (line 60).
8. “were INVITED TO PARTICIPATE in the study” (line 68).
9. The nature of the missing information (line 71) needs to be clarified.
10. Delete “have” (line 71).
11. “incomplete data” (line 72).
12. “average” is presumably “mean” (line 74) and while plus/minus notation is commonly used to indicate standard deviations, this should be defined for the first such use.
13. The number of decimal places (e.g. line 74) seems excessive to me.
14. A flowchart showing the information about the patients invited and eventually analysed (lines 66-79) would be useful.
15. Presumably, the level of significance is two-sided? (line 94)
16. “acceptable” rather than “available”? (line 103)
17. “OF A 45º line, wheRE” (line 106).
18. For R packages, the actual name should be used (line 113) and I’d suggest this be a reference with a link to CRAN or elsewhere.
19. “MEAN age” (line 117).
20. Note that while SPSS reports p-values as “0.000” in some cases, the p-value is of course not actually zero and I would recommend reporting such small p-values as “p<0.001” instead.
21. The number of diseased vessels (line 121) is presumably referring to the mean number of there? Given the discrete nature of this variable, I would have suggested Mann-Whitney-Wilcoxon instead. The number of grafts is a simialr case.
22. Line 125 seems to be present mean mortality percentage estimates rather than rates.
23. AUCs should be accompanied by CIs (e.g. lines 130-131). Note these appear to be included in the figures but are not labelled as such there.
24. “in THE elderly group IS shown in” (line 141).
25. “BOTH showed” (line 146).
26. “systemS” (line 148).
27. “for THE younger” (line 156).
28. “in THE younger group IS shown” (line 158).
29. “THE younger” (line 162).
30. “THE EuroSCORE” (line 170).
31. “world, HAVE BEEN well” (line 171).
32. Lines 181-182 are presumably about confidence in the use of the system?
33. Should “epidemiology” be something about patient characteristics, health behaviours, disease progression or similar?
34. I would move the Discussion points about characteristics of the SinoSCORE patients to the results (lines 189-193).
35. “performance” rather than “power”? (line 196)
36. “ALL had” (line 197).
37. “substantially” rather than “sensibly”? (line 198)
38. “THE elderly” (line 199).
39. “published RECENTLY, SincoSCORE” (line 228).
40. “no compromise” needs rewording (line 230).
41. For the figures, there appear to be 95% CIs in parentheses and these should be labelled as should the p-value (which I assume to be comparing all three AUCs).
42. For the tables, I would suggest “n” rather than “N” for the sample sizes.
43. Table 2 suggests the p-values are from Cox’s proportional hazards models, which doesn’t match the methods.
44. For surgery status, I am assuming this was a three-level variable and so compared with one and not three tests. The overall p-value would be more appropriate than the three pairwise comparisons.
45. Table 3’s headings could be clearer so the reader does not need to read the abbreviations.

---

## Round 0.2 · Major Revisions

· Academic Editor

Major Revisions

Dear authors,

One of the reviewers has indicated your manuscript needs some major changes before publication in PeerJ. Try to apply them in a new version of the text in order to re-evaluate your paper (MAJOR REVISION).

With respect and warm regards,
Dr Palazón-Bru (academic editor for PeerJ)

Reviewer 1 ·

Basic reporting

no comment

Experimental design

no comment

Validity of the findings

no comment

Additional comments

This revised MS could be considered to be accepted after the chages made.

·

Basic reporting

While I appreciate the authors’ point about the participants aged under 70 being a comparison group, and this would justify their choice of title for the manuscript, this needs to be made clearer to the reader in the abstract, e.g., “Patients were divided into two subsets according to their age: elderly group (age≥70) and younger group (age<70).” could be “Patients were divided into two subsets according to their age: elderly group (age≥70) with a younger group (age<70) used for comparison.” (this could also be made clearer in the methods later in the manuscript) and this should then be reflected in the amount of space the manuscript dedicates to discussing the elderly, younger, and overall results. If the younger group is used for comparison purposes, as the title and authors’ comments suggest, I would expect the results related to them to be secondary in the abstract and manuscript body. At the moment, the entire cohort results are reported, then the elderly, and finally the younger (with approximately equal space for each). If the elderly group is central, they should be reported first and then the younger age group for comparison purposes. I’m not sure that results about the entire cohort are at all meaningful if the focus is on the elderly (e.g., the overall median age of 66 doesn’t make sense to me in this context).

There are still some instances of awkward or unusual wording (e.g. in the abstract “The outcome endpoint of this study was in-hospital mortality.” could written more conventionally as “The outcome of interest in this study was in-hospital mortality.” or similar; “receiver’s” should be “receiver”, see also this typo in the methods). The methods start with “Form January…” Some careful proofreading should fix these as the intended meaning of the text is generally clear. Some typos have also crept into the manuscript with the edits, e.g. “paticipation” and “SionSCORE” in the methods. I won’t list all of the typos here.

My comment that sometimes I was not sure what particular values represent remains, e.g. “an median age of 66.00(60.00,71.00)” might be showing the 25th and 75th percentiles (note that the IQR in this case would be 11 years; this is a single number summary of the variability and not two separate numbers, see: https://en.wikipedia.org/wiki/Interquartile_range) but this is not made explicit.

It’s important that readers with a wide variety of backgrounds can understand most if not all aspects of the manuscript even if things might be obvious for those with a particular background.

By the way, I’m not sure that this information about age should be repeated in both the methods and results (I would suggest that the latter would be more appropriate, although I can see an argument that the methods refer to the database available).

Experimental design

Note that “Mann-Whitney-Wilcoxon test” is not a test “for categorical variables” (see the methods) but a test for ordinal or continuous variables that do not satisfy the assumptions required for a t-test or where a t-test is not desired for other reasons. I appreciate the addition of the level of significance to the methods, and, while pedantic, would also like it made clear that this is a two-sided level of significance (I am assuming this is the case here).

I still have to disagree with “A well-calibrated risk evaluation system gave a P-value greater than 0.05.” as Hosmer-Lemeshow’s test is looking for evidence of misfit and absence of evidence cannot be interpreted as evidence of absence (i.e., a non-significant test does not mean that there was no misfit or that the model is well calibrated, merely that there was no evidence against this hypothesis). Statement such as “SinoSCORE, EuroSCORE II and STS risk evaluation system all showed good calibration in predicting in-hospital mortality (H-L: P=0.411, P=0.113 and P=0.230, respectively)” are misinterpreting the non-significant test result as a positive test result for good calibration.

Again, being pedantic, the number of groups used in the H-L test should also be made clear in the methods.

Note also that “Discrimination (statistical accuracy) was tested by calculating the area under the receiver’s operating characteristic curve (AUC)” isn’t strictly true if the AUC is being reported in isolation—in that case the AUC merely describes an estimate of the model’s discrimination ability.

I would prefer to see false positives and false negatives given much more focus (rather than just in the DCA, which is only very briefly mentioned in the results) as I suspect these are more readily interpretable to clinicians than AUCs (while the AUC does have a precise probabilistic interpretation, this is not well understood, in my experience, among clinicians). This would also support a more health economics interpretation of the results and should broaden the usefulness of the presented results (false positives and false negatives have different implications for both patients and the health system).

A problem with the SDs in the results (e.g. “Expected mortality rate of SinoSCORE, EuroSCORE II and STS risk evaluation system for the elderly group were 1.81±3.50% (95%CI 1.54-2.08), 2.61±1.66% (95%CI 2.48-2.74) and 1.66±2.08% (95%CI 1.50-1.82), respectively.”) is that the SD suggests a 95% reference range that includes negative values. This makes it clear that the expected mortality rates are positive skewed, which is entirely expected, but then contradicts the statement in the methods that “If continuous variables satisfy the normal distribution, then variables were expressed as mean±standard deviation, else variables were expressed as median and interquartile range (IQR).”

Validity of the findings

I remain unconvinced by the statement in the abstract that “SinoSCORE, EuroSCORE II and STS risk evaluation system all achieved positive calibrations in the entire cohort and subsets.” as my interpretation of the results is that all three systems performed poorly and the authors seem to agree with that interpretation in their rebutal. While they are all statistically significantly better than chance, I would have substantial concerns about using these systems in clinical practice given their (to me) poor performance in this population. Perhaps I am misunderstanding exactly what “positive calibrations” means in this context but the next sentence is “The three risk evaluation systems showed good discrimination and calibration in the entire cohort.” and this seems at odds with my interpretation and the authors’ comments in their rebuttal as does the first sentence of the conclusion at the end of the manuscript: “The three risk evaluation systems showed good discrimination and calibration in the entire cohort.”

Additional comments

Overall, I appreciate the authors making several changes in response to some of my comments and I feel the manuscript has improved but I still have concerns about the relative focuses on the entire cohort, elderly subgroup, and younger subgroup and even stronger concerns about the poor performance of the risk evaluation systems being described in such positive terms. I feel that a reader could come away from reading such a manuscript with an overly favourable interpretation of these systems.

---

## Round 0.3 · Minor Revisions

· Academic Editor

Minor Revisions

Dear authors,

I considered that your manuscript has high standards to be published in PeerJ, but before acceptance you should correct the two typos detected by one of the reviewers.

With respect and warm regards,
Dr Palazón-Bru (academic editor for PeerJ)

·

Basic reporting

No comment

Experimental design

No comment

Validity of the findings

No comment

Additional comments

Thank you for your revisions. I just have two very minor comments to correct what appear to be typos introduced in the latest version of the manuscript.

First, in “Statistical analysis was performed by t-test for continuous variables, Mann-Whitney-Wilcoxon test and Fisher’s exact test or c2 (chi-square) test for ordinal or continuous variables that do not satisfy the assumptions required for a t-test or where a t-test is not desired for other reasons.” the reader might be confused by the order of the material and perhaps something like the following might be clearer: “Statistical analysis comparing groups was performed using t-tests for continuous variables, Mann-Whitney-Wilcoxon tests for ordinal or continuous variables that do not satisfy the assumptions required for a t-test, and Fisher’s exact or c2 (chi-square) tests for categorical variables.”

Secondly, in “The H-L statistic measured the differences between expected and observed outcomes. P-value greater than 0.05 means there is no evidence that this risk evaluation system is bad calibrated.”, I’d use “poorly” rather than “bad”, so “The H-L statistic measured the differences between expected and observed outcomes. P-value greater than 0.05 means there is no evidence that this risk evaluation system is poorly calibrated.”

---

## Round 0.4 · accepted · Accept

· Academic Editor

Accept

Dear authors,

I am happy to report that your paper has been accepted for publication in its current form in PeerJ.

Congratulations!

With respect and warm regards,
Dr Palazón-Bru (academic editor for PeerJ)

·

Basic reporting

No comment.

Experimental design

No comment.

Validity of the findings

No comment.

Additional comments

Thank you for making these minor revisions. I have no further comments.